# Genetic Ablation of the Nutrient Sensor Ogt in Endocrine Progenitors Is Dispensable for β-Cell Development but Essential for Maintenance of β-Cell Mass

**DOI:** 10.3390/biomedicines11010105

**Published:** 2022-12-30

**Authors:** Alicia Wong, Brian Akhaphong, Daniel Baumann, Emilyn U. Alejandro

**Affiliations:** 1Department of Genetics, Cell Biology, and Development, University of Minnesota Twin Cities, Minneapolis, MN 55455, USA; 2Department of Integrative Biology and Physiology, University of Minnesota Twin Cities, Minneapolis, MN 55455, USA

**Keywords:** *O*-GlcNAc, Pdx1, development, β-cells, pancreas, Pdx1, diabetes

## Abstract

Previously we utilized a murine model to demonstrate that Ogt deletion in pancreatic progenitors (OgtKO^Panc^) causes pancreatic hypoplasia, partly mediated by a reduction in the Pdx1-expressing pancreatic progenitor pool. Here, we continue to explore the role of Ogt in pancreas development by deletion of Ogt in the endocrine progenitors (OgtKO^Endo^). At birth OgtKO^Endo^, were normoglycemic and had comparable pancreas weight and α-cell, and β-cell mass to littermate controls. At postnatal day 23, OgtKO^Endo^ displayed wide ranging but generally elevated blood glucose levels, with histological analyses showing aberrant islet architecture with α-cells invading the islet core. By postnatal day 60, these mice were overtly diabetic and showed significant loss of both α-cell and β-cell mass. Together, these results highlight the indispensable role of Ogt in maintenance of β-cell mass and glucose homeostasis.

## 1. Introduction

The enzyme *O*-GlcNAc Transferase (Ogt) catalyzes the attachment of the *O*-GlcNAc post translational modification onto Serine and Threonine residues on nuclear, cytosolic proteins, and mitochondrial proteins [1,2,3,4]. This attachment is removed by the enzyme, *O*-GlcNAcase (Oga), whose namesake gene contains a diabetes susceptibility locus in humans [5]. UDP-GlcNAc, the substrate for Ogt, is synthesized through the glucose-sensitive Hexosamine Biosynthesis Pathway (HBP) [6]. Distinct from other forms of protein glycosylation, the addition and removal of an *O*-GlcNAc modification can occur many times at different rates throughout the lifespan of a single polypeptide [7] and is sensitive to changes in cellular environment, such as the influx of nutrients [8]. Aberrations in *O*-GlcNAcylation have been linked with the development of metabolic diseases such as Type 2 Diabetes [9,10,11,12], and assessment of global *O*-GlcNAcylation levels has been shown to be a potential early biomarker of insulin resistance in humans [13].

While expressed in all mammalian tissues, Ogt is most abundant in the pancreas [14], and within the adult murine model, it is mostly highly expressed in the pancreatic islets [15]. Pancreas development is tightly governed by a hierarchy of transcriptional activation. Starting in the foregut endoderm, a pool of Pdx1^+^ pancreatic progenitors gives rise to both the endocrine (Islets of Langerhans) and exocrine (acinar) pancreas compartments [16]. In the mature pancreas, Pdx1 is only expressed in the β-cells [17]. Some transcription factors are *O*-GlcNAc modified by Ogt [18], including Pdx1 [19,20]. Previously, we demonstrated that *O*-GlcNAcylation in the pancreatic epithelium is required for pancreas organogenesis in mice. Deletion of Ogt in the developing epithelium resulted in pancreatic hypoplasia due, in part, to increased apoptosis and a reduction in the Pdx1^+^ pancreatic progenitor pool [20]. In the β-cells, ablation of Ogt led to reduced β-cell mass and increased ER stress-mediated apoptosis [15]. Using both murine and human islets, we previously demonstrated that *O*-GlcNAcylation of sarco/endoplasmic reticulum Ca^2+^-ATPase, SERCA2, in the islet is required for lipid potentiation of insulin secretion, a hallmark of β-cell adaptation to hyperlipidemia and obesity prior to the onset of Type 2 Diabetes [21]. Many transcription factors that play an important role in β-cell function in response to glucose stimulus, such as cAMP-responsive element-binding protein (CREB), and Pdx1, have been demonstrated to be *O*-GlcNAc modified, and the modification is known to affects their activity [19,22]. *O*-GlcNAcylation of several components of the insulin signaling pathway, such as insulin receptor substrate 1 (IRS-1), has been shown to attenuate insulin signaling under a normoglycemic context [23]. On the other hand, hyper-*O*-GlcNAcylation in the β-cells also results in β-cell dysfunction [24]. Taken together, control of *O*-GlcNAcylation levels is important for pancreas development and the maintenance of β-cell function, adaptation, and glucose homeostasis. We hypothesized that the *O*-GlcNAc modification of transcription factors at the endocrine development stage is required for the formation of functional pancreatic islets.

Using an endocrine-cell Ogt-deficient mouse model (OgtKO^Endo^), we demonstrate that Ogt is not required for the formation of pancreatic islets. However, OgtKO^Endo^ develop aberrant islet architecture and progressive hyperglycemia early in life, followed by severe loss of α-cell mass and β-cell mass in adulthood. Taken together, these data highlight the requirement of Ogt in the maintenance of functional β-cell mass.

## 2. Methods

### 2.1. Animals

Neurogenin3-cre (Ngn3), Ogt^flox/flox^, and CAG-EGFP reporter transgenic animals were purchased from The Jackson Laboratory. Ngn3-cre; Ogt^f/+^ (OgtHET^Endo^) females were crossed with Ogt^f/y^ males to produce OgtHET^Endo^, OgtKO^Endo^, and controls (Ogt^flox/+^, Ogt^flox/flox^, Ogt^flox/y^, or Ngn3-cre; Ogt WT). All mice were generated on a C57Bl6/J background and group-housed on a 14:10 light:dark cycle. Experiments were performed with both male and female mice (unless otherwise stated), and data was combined for neonatal experiments and separated in adolescent and adult experiments.

### 2.2. Blood and Tissue Collection

Neonatal mice were sacrificed by decapitation at postnatal day 0. P23, p28, and p60 mice were sacrificed by CO_2_ asphyxiation, and death was confirmed via a hard toe pinch followed by cervical dislocation. Pancreas and spleen were collected following a V-incision to open the abdominal cavity then microdissected under a gross morphology microscope. Whole blood and blood glucose readings were collected via the trunk at sacrifice for neonates and via tail vein for p23, p28, and p60 mice.

### 2.3. Serum Insulin Analysis

Non-fasting serum insulin levels were assessed using Rat or Mouse Ultrasensitive Insulin ELISA kit (Alpco Diagnostics, Salem, NH, USA) using mouse insulin standards and following manufacturer’s instructions. Serum was extracted from whole blood collected at sacrifice.

### 2.4. Glucose Tolerance Test

Mice were fasted overnight for 14 h followed by an intraperitoneal injection of sterile glucose at a concentration of 2 g/kg body weight. Blood glucose readings were assessed prior to injection and at 30, 60, and 120 min post-injection via tail vein using a handheld glucometer, which has an upper limit of 600 mg/dL. Any values above 600 mg/dL were calculated using blood diluted 1/5.

### 2.5. Tissue Preparation and Immunofluorescence Staining

Tissues harvested at dissection were treated in 3.7% formaldehyde for 12 h at 4 degrees C, rehydrated in 70% ethanol, and processed for paraffin embedding using the Leica ASP300S. Neonatal tissue was sectioned from top to bottom at 5 μm thickness. Tissues from p23, p28, and p60 were sectioned at 5 microns thickness representing 5 regions spaced 200 μm apart as previously described [25]. Deparaffinization was performed via Citrisolv and ethanol baths, and heat-based antigen retrieval was conducted 3 × 4 min at 95 degrees C using a sodium citrate solution. Tissues were permeabilized via TritonX-100 in PBS, followed by blocking using Roche Blocking Reagent (Sigma, St. Louis, MO, USA). Tissues were stained using primary antibodies against guinea pig insulin (DAKO, Santa Clara, CA, USA, 1:400), mouse glucagon (Abcam, Cambridge, UK, 1:500), rabbit Pdx1 (Abcam, 1:400), and rabbit synaptophysin (Abcam, 1:400) and incubated overnight at 4 degrees C followed by treatment with FitC (1:400), Cy3 (1:500), or Alexa647 (1:100)-conjugated secondary antibodies. Nuclei were counterstained by dipping in DAPI (1:1000 in PBS). TUNEL apoptotic staining was completed using the ApopTag Red in situ Apoptosis Detection Kit per manufacturer’s instructions (Millipore, Burlington, MA, USA).

### 2.6. Cell Counts and Morphometric Analysis

All cell counts and morphometric analysis were performed using FIJI/ImageJ [26] using the thresholding, masking, and object counter as previously described [20]. Fluorescence images were captured using the Nikon Eclipse NI-E epifluorescence microscope (Nikon Instruments, Melville, NY, USA) at 4× (gross morphology), 10× (β-cell and α-cell area and mass), 20× (islet morphology), and 40× (islet cell counts). All cell counts represent at least 1000 cells per mouse. Cell type was identified using hormone^+^ area by immunofluorescence, which was used to quantify α-cell area, β-cell area, and pancreas area in FIJI/ImageJ [26]. α and β-cell mass was conducted by averaging insulin^+^ area or glucagon^+^ area over pancreas area across five sections throughout the pancreas, then correcting this global average by pancreas weight.

### 2.7. Statistical Analysis

Statistical analysis was conducted in Prism 8.4.3 for windows (GraphPad Software, San Diego, CA, USA, www.graphpad.com (accessed on 7 December 2022)) via Mann–Whitney U-Test with *p* ≤ 0.05 deemed significant and 0.05 < *p* < 0.1 deemed trending. All values are expressed as mean ± SEM.

## 3. Results

### 3.1. Deletion of Ogt in the Endocrine Progenitors Has No Effect on Islet Formation

To assess the effect of Ogt on the formation of the islet cells, we used a cre-lox system driven by the Neurogenin-3 (Ngn3) promoter, to genetically ablate Ogt in all cells of neuroendocrine origin (OgtKO^Endo^). We assessed these mice at birth (postnatal day (*p*) 0). Both heterozygous and homozygous deletion of Ogt resulted in no differences in gross pancreas morphology (Figure 1A). OgtKO^Endo^ displayed similar body weight, pancreas weight, and blood glucose as littermate controls (Figure 1B–D), along with comparable β-cell mass and α-cell mass (Figure 1E,F). These parameters were also comparable between OgtHET^Endo^ and controls (Appendix A). Both islet morphology (and pancreas morphology were normal in OgtKO^Endo^ (Figure 1G and Appendix A). Because programmed cell death is involved in the remodeling of neonatal islets [27], we also assessed apoptosis in the newborn islets via immunofluorescence. Staining by TUNEL showed similarly low levels of apoptosis in the β-cells of OgtKO^Endo^ and control (Figure 1H,I). Endogenous GFP cre-reporter expression was robust and specific to pancreatic endocrine cells, indicating high efficiency of the Ngn3-cre (Figure 1J). Taken together, these data indicate that Ogt is not required for the formation of endocrine cells.

### 3.2. Progressive Loss of β-Cell Mass and Reduced Immunoreactivity of Pdx1 in OgtKO^Endo^

To determine the effect of Ogt in the maintenance of glucose homeostasis, we tracked various parameters of OgtKO^Endo^ from p15 through p28. At p15-17, both male and female OgtKO^Endo^ exhibit comparable body weight and non-fasting blood glucose to littermate controls, though OgtKO^Endo^ females showed a non-significant trend toward lower blood glucose (Figure 2A,B and Appendix A
*p* = 0.0952). At p28, male OgtKO^Endo^ show similar body weight, absolute pancreas weight, and pancreas/body weight ratio compared to littermate controls (Figure 2C–E). However, female OgtKO^Endo^ exhibit a non-significant trend toward decreased body weight at this age (Appendix A
*p* = 0.0686). In line with this, while absolute pancreas weight was comparable, pancreas weight corrected over body weight was reduced in female OgtKO^Endo^ (Appendix A). Both male and female OgtKO^Endo^ have elevated non-fasting blood glucose at p28 but have comparable serum insulin levels (Figure 2F,G and Appendix A). Since blood glucose and serum insulin levels were similar between male and female at p28, we performed histological analyses in male mice only. OgtKO^Endo^ have reduced insulin-positive area relative to whole pancreas area (Figure 2H) and a non-significant trend toward reduced β-cell mass (Figure 2I *p* = 0.0556). In contrast to β-cell specific Ogt-deficient mice, OgtKO^Endo^ mice also exhibited trends in reduced glucagon^+^ area and α-cell mass (Figure 2J *p* = 0.0635, K *p* = 0.1111). Because Pdx1 expression in β-cells is positively correlated with function and survival, we assessed Pdx1 protein expression via immunofluorescence. Staining with Pdx1 antibody suggested reduced immunoreactivity of Pdx1 in the OgtKO^Endo^ (Figure 2L). Pdx1 exerts its activity as a transcription factor primary in the nucleus. Interestingly, in islets of OgtKO^Endo^ mice, some cells express Pdx1 in the cytoplasm (Figure 2L).

### 3.3. Aberrations in Islet Architecture in OgtKOEndo at Weaning

Because OgtKO^Endo^ are normoglycemic at p15-17 by hyperglycemic at p28, we assessed these mice at the intermediate age of p23 to assess the presence of transitional changes within the islet. At p22–24, both male and female OgtKO^Endo^ have similar body weight, absolute pancreas weight, and pancreas weight/body weight ratio as controls (Figure 3A–C and Appendix A). However, both male and female OgtKO^Endo^ display elevated, albeit more variable, non-fasting blood glucose levels (Figure 3D and Appendix A), with comparable but more variable serum insulin levels (Figure 3E and Appendix A). Since both male and female mice were metabolically similar at this age, we performed histological analyses only in male mice. Both insulin^+^ area /pancreas area and β-cell mass were comparable between OgtKO^Endo^ and littermate controls (Figure 3F,G). Interestingly, while glucagon^+^ area/pancreas area and α-cell mass were not different between OgtKO^Endo^ and control (Figure 3H,I), the proportion of glucagon^+^ cells per islet was reduced (Figure 3J). Islet architecture in these mice was also perturbed, with α-cells invading the islet core (Figure 3K), but exocrine pancreas morphology was unchanged (Appendix A).

### 3.4. Adult OgtKOEndo Are Diabetic and Have Both Reduced β-Cell Mass and α-Cell Mass

Sex hormones are known to play a role in regulating glucose homeostasis. Thus, we wanted to assess OgtKO^Endo^ mice at sexual maturity. We first confirmed robust expression of Ngn3-cre in adults via visualization of GFP cre-reporter expression in p60 mice and showed that cre expression colocalized with both α-cells and the β-cells (Figure 4A). No differences in exocrine morphology were observed between OgtKO^Endo^ and controls (Appendix A), and pancreas weight and spleen weight were comparable to controls (Figure 4B,C). Both male and female OgtKO^Endo^ displayed severely elevated blood glucose levels (Figure 4D and Appendix A), with a corresponding decrease in body weight over time (Figure 4E). Both male and female OgtKO^Endo^ were severely glucose intolerant and unable to normalize blood glucose levels following in intraperitoneal injection of glucose (Figure 4F and Appendix A). Because no differences in glucose homeostasis were observed between males and females, histological analyses were performed in male mice only. As expected, both β-cell mass and α-cell mass were severely reduced in OgtKO^Endo^ (Figure 4G–I). In instances of massive β-cell mass loss, other islet cells such as the somatostatin-producing δ-cells have been shown to transdifferentiate into β-like cells in compensation [28]. To assess whether this was occurring in the OgtKO^Endo^, we performed immunofluorescence staining of pancreas sections using synaptophysin, a marker of cells of neuroendocrine origin. We did not find a substantial population of synaptophysin-expressing cells outside of the insulin and glucagon positive islets in the OgtKO^Endo^ (Appendix A).

## 4. Discussion

We have previously shown the requirement of Ogt in the developing epithelium in pancreas formation, but the role of Ogt in development of the pancreatic endocrine cells was unknown [20]. Here, we genetically ablated Ogt in the pancreatic endocrine progenitors and characterized the morphological and metabolic phenotypes using mice. We found that while Ogt is not required for formation of the endocrine pancreas at birth, it is indispensable for the maintenance of both islet architecture and β-cell and α-cell mass and, consequently, glucose homeostasis (Figure 5).

### 4.1. Role of Islet Protein O-GlcNAcylation in Glucose Homeostasis

In the pancreas, nutrient-sensitive *O*-GlcNAcylation occurs at a relatively higher level in the islets than in the acinar [15]. In the β-cell, both hypo and hyper-*O*-GlcNAcylation increases susceptibility to metabolic stress and β-cell dysfunction [15,24,29], suggesting modulation of *O*-GlcNAc levels is necessary to maintain β-cell function. In the α-cells, deletion of Ogt leads to defects in both glucagon secretion and gluconeogenesis but does not affect whole body glucose homeostasis as measured by glucose tolerance or insulin sensitivity tests, and fasting glucose and insulin levels [30]. Deletion of Ogt in β-cells or α-cells do not impair peripheral insulin sensitivity as measured by insulin tolerance test via ip injection [15,30]. In the present study, Ogt was simultaneously deleted in all endocrine cells, including both α-cells and β-cells. It is known that β-cell-specific ablation of Ogt leads to hyperglycemia in male mice at approximately 10 weeks of age and females at 15 weeks [15]. In this study, however, hyperglycemia occurs in both males and females by p23 (3 weeks of age), indicating that disruption of *O*-GlcNAcylation in multiple cell types at once accelerates the development of hyperglycemia. It is known that different islet cell types communicate via paracrine signaling to sustain normoglycemia [31]. For example, insulin signaling in the α-cells has been shown to regulate glucagon secretion [32]. Likewise, blocking the glucagon receptors in transplanted human islets in mice results in increased blood glucose levels and reduced insulin levels in vivo [31], and treatment of human islets with glucagon strongly potentiates glucose-stimulated insulin secretion in vitro [33]. It is surmisable that concurrently deleting Ogt in multiple islet cell types perturbs cell to cell communication and leads to swift defects in function.

### 4.2. Maintenance of the α-Cell and β-Cell Population

Ablation of Ogt in either the β-cells or the α-cells results in reduced β-cell mass and α-cell mass, respectively [15,20]. In the β-cell Ogt deletion model, preliminary data at one moth of age suggests α-cell mass was not perturbed (data not shown). This is in line with evidence indicating β-cells and α-cells differentiate from separate cell lineages in adults [34], suggesting perturbation of one cell type, specifically, does not occur at the detriment of the other. In the present study, we demonstrate that simultaneous deletion of Ogt in all endocrine cells results in progressive loss of both β-cell mass and α-cell mass without upregulation of other islet cell types. This suggests Ogt may also be required in the functional mass of other pancreatic endocrine cell types and is an interesting direction for future study.

### 4.3. Pro-Survival Role of Ogt in the Endocrine Cells

Pdx1 is a master regulator of β-cell development, identity, and function [35]. In the postnatal pancreas, Pdx1 expression is restricted to the β-cells [17]. Expression of Pdx1 potentiates transcription of the insulin gene and represses an α-cell program [36,37]. Depletion of Pdx1 results in increased apoptosis in the islet, demonstrating the pro-survival role of Pdx1 [38]. Interestingly, we and others have demonstrated that Pdx1 is *O*-GlcNAc modified by Ogt [19,20] and that mice with Ogt-deficiency in the β-cells have reduced Pdx1 protein levels and increased ER stress-mediated apoptosis [15], suggesting Ogt regulates cell survival through Pdx1. Reconstitution of Pdx1 partially improves pancreatic and β-cell mass deficits in mice lacking Ogt in their pancreatic progenitors [39]. In the present study, we demonstrate that at p28, OgtKO^Endo^ mice displayed reduced immunoreactivity to Pdx1. These mice go on to become severely glucose intolerant, with near-complete loss of β-cell mass, suggesting Pdx1-dependent apoptosis is a potential mechanism of the progressive β-cell mass and serum insulin reduction in the OgtKO^Endo^. The pool of Pdx1^+^ cells is also a determinant of the population of other islet cell types [40]. Mice with β-cell specific Pdx1-deficiency display aberrant islet architecture [40]. In the present study, islets of OgtKO^Endo^ mice also show differences in islet architecture, with α-cells invading the core of the islet (a common phenotype in diabetic islets), suggesting that in the progression toward islet cell loss, deletion of Ogt perturbs islet organization, in part, by reduction of Pdx1. Islet apoptosis and Pdx1 protein levels are potential areas of exploration in a future study.

### 4.4. Temporal Role of Ogt in Regulation of Pancreas Development

We previously demonstrated that expression of Ogt at the pancreatic progenitor stage (OgtKO^Panc^) is necessary for both endocrine and exocrine pancreas development. Ablation of Ogt at this stage results in severe pancreatic hypoplasia [20]. In the present study, Ogt was deleted in the endocrine progenitors, which arise later in development (OgtKO^Endo^). Unlike OgtKO^Panc^ mice, OgtKO^Endo^ are born with normal exocrine and endocrine pancreas compartments, like the β-cell-specific or α-cell specific Ogt knockout [15,30]. This suggest that Ogt plays a crucial role in pancreas organogenesis prior to endocrine specification, after which Ogt is dispensable. This temporal nature of Ogt’s regulation in pancreas development may point to the critical windows in development during which O-GlcNAcylation of transcription factors is essential for progenitor cell survival (Figure 5).

### 4.5. Limitations of the Study

We cannot rule out the effects on glucose metabolism of Ogt deletion in extra-pancreatic tissues (i.e., hypothalamus) where the Ngn3 Cre driver may also be expressed [41]. Thus, one limitation of the current study is the expression of the Ngn3 promoter in hypothalamic neurons [42] and in the duodenum [43], where Ogt loss has been associated with obesity and reduced body weight in adult mice, respectively. We monitored body weight in our studies, and we have not observed any changes in body weight phenotypes in OgtKO^Ins^, OgtKO^Endo^, or OgtKO^Panc^ that cannot be attributed to the onset of hyperglycemia. Additionally, this study is a preliminary, preclinical assessment on pancreas development and the programming of islet susceptibility to Type 2 Diabetes with limited animal cohorts. The effect of nutrient-sensitive post-translational modifications of transcription factors in the developing human exocrine and endocrine pancreas warrants further exploration.

### 4.6. Speculation on Clinical Utility of Present Data

An individual’s susceptibility to glucose intolerance is determined by genetic, epigenetic, and environmental factors. Genome Wide Association Studies (GWAS) have identified genetic loci associated with β-cell function and glucose-stimulated insulin release as Type 2 Diabetes susceptibility alleles [44]. This suggests that β-cells’ ability to functionally compensate in response to insulin demand is a tipping point in determining whether an individual will develop Type 2 Diabetes. Obesity is a major risk factor in developing Type 2 Diabetes [45]. Human islets from male obese BMI donors exhibit higher insulin secretion in response to glucose stimulation when there is higher islet *O*-GlcNAcylation while lower islet *O*-GlcNAcylation is associated with diminished islet stimulation index compared to a lean BMI donor control [21], suggesting upregulation of islet *O*-GlcNAcylation is a potential functional adaptation to increased insulin demand associated with hyperlipidemia.

During pancreas development, the availability of nutrients in the intrauterine environment is known to program offspring metabolic health and susceptibility to obesity and metabolic syndromes [46] but also influences flux through the HBP and, consequently, levels of O-GlcNAcylation. Many transcription factors that regulate pancreas development and the maintenance of β-cell identity and function, such as Pdx1, are putatively *O*-GlcNAc modified [19]. This is a potential mechanistic link between nutrient-dependent post-translational modifications and the programming of offspring metabolic function, an avenue that warrants further study. In addition, while hemoglobin A1c (HbA1c) test is a common test to diagnose prediabetes and diabetes, measuring levels of *O*-GlcNAc-modified proteins in the blood has recently emerged as a more sensitive biomarker of early-stage metabolic dysfunction in young, metabolically normal adults and was positively correlated with insulin resistance (HOMA-IR) [13]. Thus, measurement of *O*-GlcNAc levels in conjunction with other standard clinical tests such as glucose tolerance and HbA1cc tests may be an avenue to explore to detect individuals with early onset of the disease to maximize effective treatment or prevention.

## 5. Concluding Remarks

While it is known that deletion of Ogt in the pancreatic β-cells or α-cells leads to defects in β-cell and α-cell function, respectively, the effect of simultaneous ablation of Ogt in all pancreatic endocrine cells has not been reported. We demonstrate that while Ogt expression in the pancreatic endocrine progenitors is not required for the formation of the islets at birth, it is essential for the regulation of glucose homeostasis through maintenance of functional α-cell and β-cell mass, highlighting the temporal role of Ogt in regulating pancreas development (Figure 5).

## Figures and Tables

**Figure 1 biomedicines-11-00105-f001:**
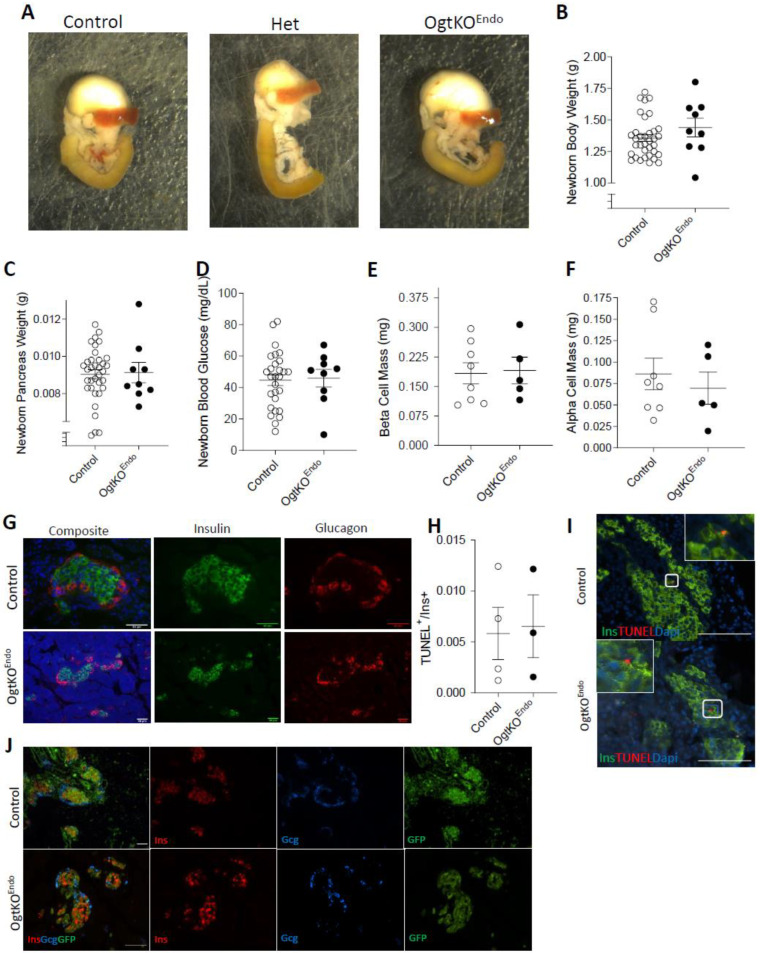
No morphological or metabolic changes in OgtKO^Endo^ at birth. At p0: Gross morphology of pancreas, stomach, spleen, and duodenum (**A**), body weight (**B**, *n* = 9–24), pancreas weight (**C**, *n* = 9–24), and random blood glucose (**D**, *n* = 9–24). Quantification of β-cell mass (**E**, *n* = 5–8) and α-cell mass (**F**, *n* = 5–8). Representative immunofluorescence images showing normal islet morphology (**G**, Top row 40×, bottom row 20×, scale = 50 μm). Quantification of apoptosis in β-cells via TUNEL (**H**, *n* = 3) with representative images (**I**, 40×, scale = 100 μm). CAG-EGFP reporter indicates Cre-expression in both β and α-cells (**J**, 20×, scale = 50 μm). Statistics were performed with a Mann–Whitney U-Test (two-tailed with significance *p* ≤ 0.05). In all graphs, open circle = Control and closed circle = OgtKO^Endo^. Both male and female data combined.

**Figure 2 biomedicines-11-00105-f002:**
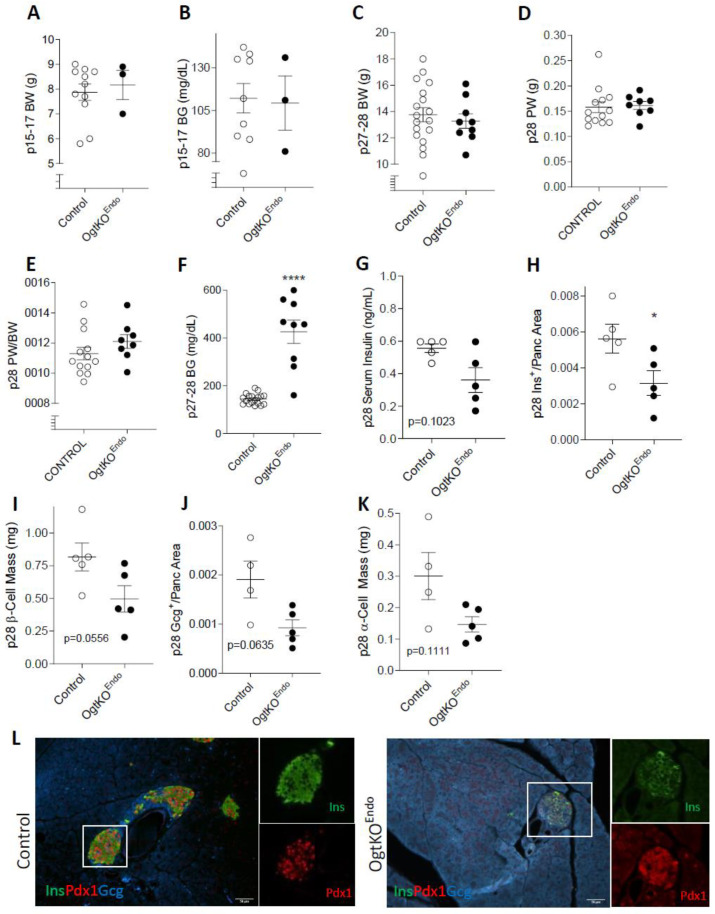
Progressive hyperglycemia and reduced Pdx1 Immunoreactivity in male OgtKO^Endo^. Body weight (**A**, *n* = 3–11) and random blood glucose (**B**, *n* = 3–11) at p15-17. Body weight (**C**, *n* = 9–18), pancreas weight (**D**, *n* = 8–13), and pancreas weight corrected over body weight (**E**, *n* = 8–13, *p* = 0.1403) at p28. Non-fasting blood glucose levels in OgtKO^Endo^ (**F**, *n* = 9–18, **** *p* < 0.0001) and serum serum insulin (**G**, *n* = 5, *p* = 0.1023) at p28. Quantification of Ins^+^ area/pancreas area (**H**, *n* = 5, * *p* = 0.0317), β-cell mass (**I**, *n* = 5, *p* = 0.0556), Glucagon^+^ area/pancreas area (**J**, *n* = 4–5, *p* = 0.0635), and α-cell mass (**K**, *n* = 4–5, *p* = 0.1111) at p28. Representative immunofluorescence images of cytoplasmic Pdx1 expression in OgtKO^Endo^ (**L**, 20×, scale = 50 μm). Statistics were performed with a Mann–Whitney U-Test (two-tailed with significance *p* ≤ 0.05). In all graphs, open circle = Control and closed circle = OgtKO^Endo^. Female data in Appendix A.

**Figure 3 biomedicines-11-00105-f003:**
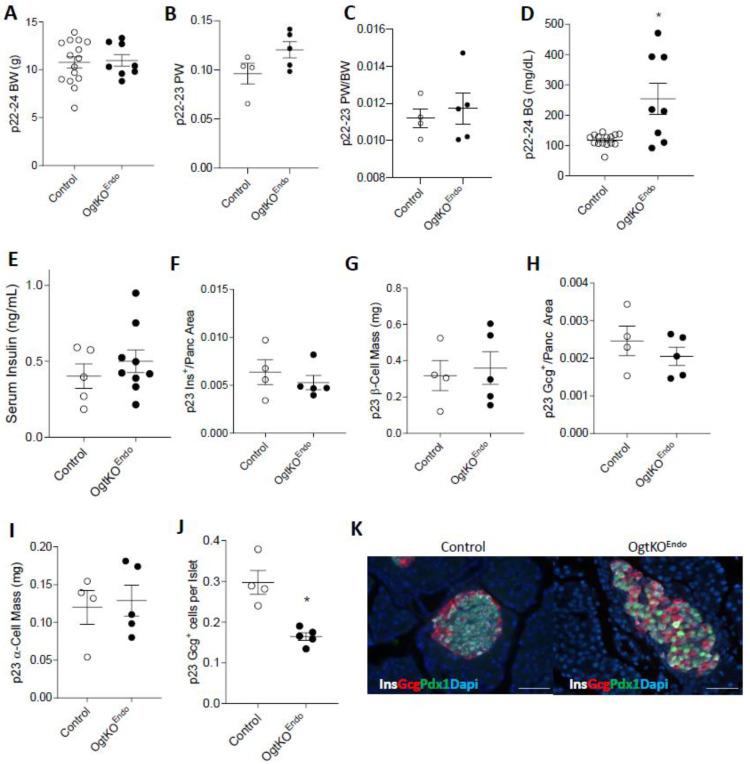
Disregulated islet architecture in male p23 OgtKO^Endo^ mice. At p23: Bodyweight (**A**, *n* = 8–15), pancreas weight (**B**, *n* = 4–6), and pancreas weight corrected over bodyweight (**C**, *n* = 4–5). Elevated non-fasting blood glucose (**D**, *n* = 8–15, * *p* = 0.0151) in OgtKO^Endo^. Serum insulin levels from non-hyperglycemic (BG < 150 mg/dL) OgtKO^Endo^ (**E**. *n* = 5–9). Quantification of Insulin^+^/Pancreas area (**F**, *n* = 4–5), β-cell mass (**G**, *n* = 4–5), Glucagon^+^/Pancreas area (**H**, *n* = 4–5), α-cell mass (**I**, *n* = 4–5), and Glucagon^+^ cells/islet (**J**, *n* = 4–5, * *p* = 0.0159, quantified over total number of Ins^+^, Gcg^+^, and Ins^+^/Gcg^+^ cells). Representative images of the islet showing α-cell invasion to the islet core in OgtKO^Endo^ (**K**, 40×, scale = 50μm). Statistics were performed with a Mann–Whitney U-Test (two-tailed with significance *p* ≤ 0.05). In all graphs, open circle = Control and closed circle = OgtKO^Endo^. Female data in Appendix A.

**Figure 4 biomedicines-11-00105-f004:**
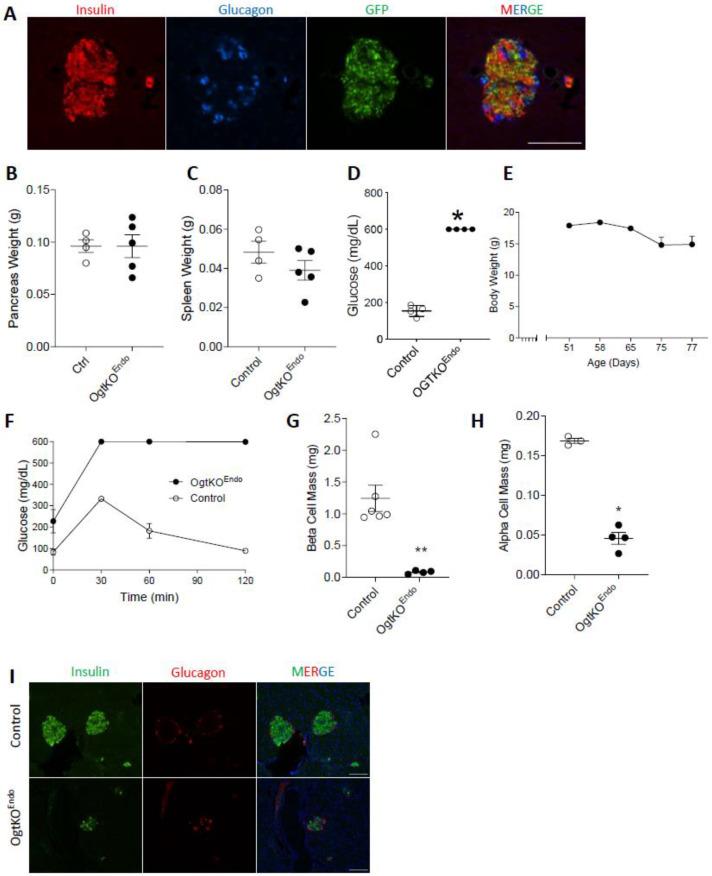
Severe glucose intolerance and loss of β-cell mass in male OgtKO^Endo^. Representative image of OgtKO^Endo^ at p137 showing robust GFP cre-reporter expression (**A**, 20×, scale = 100 μm). Pancreas weight (**B**), and spleen weight (**C**) at p60. Non-fasting blood glucose levels (**D**), body weight progression (**E**), and glucose tolerance test (**F**) at p60. Quantification of β-cell mass (**G**) and α-cell mass (**H**) showing severe reduction in OgtKO^Endo^ at p60. Representative immunofluorescence images of islets at p60 (**I**, 20×, scale = 100 μm). Statistics were performed with a Mann–Whitney U-Test (two-tailed with significance *p* ≤ 0.05; * *p* ≤ 0.05, ** *p* ≤ 0.01). In all graphs, open circle = Control and closed circle = OgtKO^Endo^.

**Figure 5 biomedicines-11-00105-f005:**
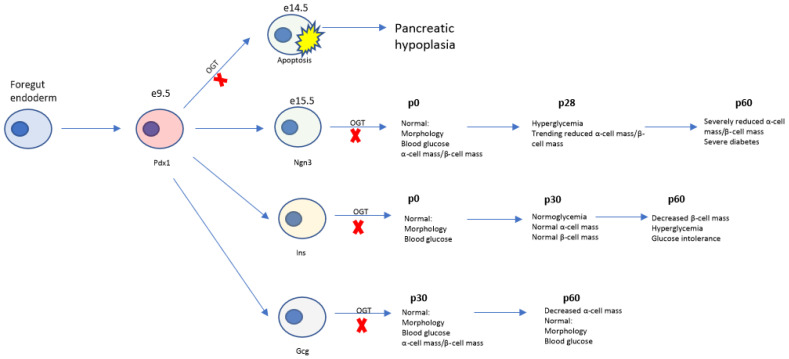
Summary of Ogt loss at various timepoints in pancreas development. Pancreas development begins in the foregut endoderm. A pool of Pdx1^+^ pancreatic progenitor cells give rise to both the endocrine and exocrine pancres. Deletion of Ogt at the pancreatic progenitor stage leads to pancreatic hypoplasia, in contrast to other models in which Ogt is deleted later in development.

## Data Availability

Data available at request.

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
