# Peer review of "Genetic Ablation of the Nutrient Sensor Ogt in Endocrine Progenitors Is Dispensable for β-Cell Development but Essential for Maintenance of β-Cell Mass"

_biomedicines, 2022, doi:10.3390/biomedicines11010105_

Round 1

Reviewer 1 Report

The manuscript entitled “Genetic Ablation of the nutrient-sensor Ogt in endocrine progenitors is dispensable for β-cell development but essential for maintenance of β-cell mass”. This study underlines the the indispensable role of Ogt in maintenance of β-cell mass and glucose homeostasis. It needs some revisions before acceptance.

Comments

The authors need to measure the metabolic parameters in mice as well as protein expression related to insulin sensitivity i.e.  AKT, AMPK, PGC1 alpha and  Insulin receptor.

Author Response

We thank the reviewer for this suggestion. In the present study, due to early development of overt-diabetes early in life with massive loss of islet cell mass when Ogt is deleted in pancreatic endocrine cells, we could not isolate enough islets from these mice to obtain enough protein for Western Blot to assess Akt or AMPK as suggested. However, in our previous study where Ogt was specifically deleted in pancreatic beta-cells, insulin sensitivity was normal (Alejandro et al, Cell Reports, 2015, Lockridge et al, Cell Reports, 2020). Deletion of Ogt in alpha-cells also rendered normal insulin sensitivity in peripheral tissues (Essawy, et al, JBC, 2021). In the islets of Ogt deficient beta-cells, we reported reduced Akt signaling. We have now modified the discussion section to include these previous findings.

Reviewer 2 Report

Interesting and well performed study but I miss the clinical significance of these findings. Also, the authors tend to over-emphasize their results.

Major points

1. The authors over-emphasize that the results of their very preliminary study highlight the indispensable role of Ogt in maintenance of β-cell mass and glucose homeostasis. Please significantly expand the limitations of the present study, such as the preclinical design as well as the limited cohort of models included.

2. The authors need to discuss in details what is the clinical utility of the present findings. As such, it seems only a speculation performed in animal models, without any clinical utility for patients with insulin resistance or with diabetes, as well as for subjects with normoglycemia. Please add a new sub-section in the Discussion, before the Conclusions. This new sub-section should be entitled “Speculation on the clinical utility of present data”. This is mandatory for a preclinical study.

Author Response

Please significantly expand the limitations of the present study, such as the preclinical design as well as the limited cohort of models included.

We thank the reviewer for this comment. We agree that this study is preclinical and have included more clarification on the preclinical nature of this study design in the limitations section of the manuscript discussion.

The authors need to discuss in detail what is the clinical utility of the present findings. As such, it seems only a speculation performed in animal models, without any clinical utility for patients with insulin resistance or with diabetes, as well as for subjects with normoglycemia. Please add a new sub-section in the Discussion before the Conclusions. This new sub-section should be entitled “Speculation on the clinical utility of present data”. This is mandatory for a preclinical study.

We thank the reviewer for this suggestion. We have now added this section to the Discussion section, in which we speculate on islet O-GlcNAcylation as a compensatory mechanism to potentiate insulin secretion, the role of nutrients in programming susceptibility to developing Type 2 Diabetes, and the clinical utility of measuring O-GlcNAc levels in the blood as an early diagnostic measure for insulin resistance and Type 2 Diabetes risk.

Round 2

Reviewer 2 Report

not improved.